# Validation of a Commercial Collar-Based Sensor for Monitoring Eating and Ruminating Behaviour of Dairy Cows

**DOI:** 10.3390/ani11102852

**Published:** 2021-09-29

**Authors:** Lorenzo Leso, Valentina Becciolini, Giuseppe Rossi, Stefano Camiciottoli, Matteo Barbari

**Affiliations:** Department of Agriculture, Food, Environment and Forestry, University of Florence, Via San Bonaventura, 13, 50145 Florence, Italy; lorenzo.leso@unifi.it (L.L.); giuseppe.rossi@unifi.it (G.R.); stefano.camiciottoli@unifi.it (S.C.); matteo.barbari@unifi.it (M.B.)

**Keywords:** dairy cows, sensor, collar, behavior, precision livestock farming, validation

## Abstract

**Simple Summary:**

This paper aims to validate a recently released commercial sensor (AFICollar^®^, Afimilk, Israel) to monitor dairy cows’ feeding and ruminating behavior. In order to evaluate the sensor’s performance under different feeding scenarios, the cows involved were divided into groups and fed different types of feed, including grazed pasture. Further, two version of the software used to convert raw data into behavioral information were tested and compared. Sensor data have been validated against visual observations, which served as the gold standard. Various statistical methods have been employed to assess sensor precision and accuracy. The results indicate that the sensor is adequately accurate for both feeding and ruminating time. However, the precision of the system appeared somewhat limited and should be improved.

**Abstract:**

The use of sensor technologies to monitor cows’ behavior is becoming commonplace in the context of dairy production. This study aimed at validating a commercial collar-based sensor system, the AFICollar^®^ (Afimilk, Kibbutz Afikim, Israel), designed to monitor dairy cattle feeding and ruminating behavior. Additionally, the performances of two versions of the software for behavior classification, the current software AFIfarm^®^ 5.4 and the updated version AFIfarm^®^ 5.5, were compared. The study involved twenty Holstein-Friesian cows fitted with the collars. To evaluate the sensor performance under different feeding scenarios, the animals were divided into four groups and fed three different types of feed (total mixed ration, long hay, animals allowed to graze). Recordings of hourly rumination and feeding time produced by the sensor were compared with visual observation by scan sampling at 1 minute intervals using Spearman correlation, concordance correlation coefficient (CCC), Bland–Altman plots and linear mixed models for assessing the precision and accuracy of the system. The analyses confirmed that the updated software version V5.5 produced better detection performance than the current V5.4. The updated software version produced high correlations between visual observations and data recorded by the sensor for both feeding (r = 0.85, CCC = 0.86) and rumination (r = 0.83, CCC = 0.86). However, the limits of agreement for both behaviors remained quite wide (feeding: −19.60 min/h, 17.46 min/h; rumination: −15.80 min/h, 15.00 min/h). Type of feed did not produce significant effects on the agreement between visual observations and sensor recordings. Overall, the results indicate that the system can provide farmers with adequately accurate data on feeding and rumination time, and can be used to support herd management decisions. Despite all this, the precision of the system remained relatively limited, and should be improved with further developments in the classification algorithm.

## 1. Introduction

Precision livestock farming (PLF) developed as a tool serving both farmers and researchers that has the potential to improve animal health and welfare, increase the environmental sustainability of animal farming, increase production efficiency and reduce costs [1,2]. The success of PLF in dairy farming relies on collecting meaningful data, with emphasis on individual animals. Monitoring tools need to be both easy to use for farmers and sufficiently precise for herd management [3]. A central problem is the process of the creation and acquisition of information, which allows us to convert data into informed management decisions [4]. This implies that the accuracy of data generation and manipulation must be investigated whenever new tools are implemented. This stage is necessary to evaluate the possible benefits and drawbacks deriving from the application of specific PLF tools.

Nowadays, the use of individual sensors is becoming commonplace in dairy farming, as they meet the need for automation in monitoring cows’ behavior, often with sustainable costs for the farmer. Commercial devices offer a wide range of sensors allowing the collection of a large amount of data. They include RFID (radio frequency identification) sensors on collars or ear tags to monitor the presence of the cows at the feeder [5,6], boluses [7] and systems using microphones to measure rumen activity and temperature, vision-based products for body condition score detection [8] and body temperature detection [9], as well as accelerometer-based technologies [10,11,12]. The latter are the most common [2], are offered with different attachment solutions (i.e., collar, ear, leg) and are designed to monitor the animal’s activity (active and non-active behaviors), as well as specific behaviors (feeding, drinking, rumination). Collar-based accelerometers are the most frequently marketed; however, validation studies on these specific models are less frequent than those performed on ear tags or pedometers, and therefore, additional studies are required on this specific segment [2].

The advantages of using sensors as an alternative to direct observations are related to the limited effort required for the collection of activity data while minimizing human interference and the subjective interpretation of behaviors. The use of sensors enables farmers to monitor continuously the behavior of dairy cows, to recognize variations in the animal’s activity, and thus to infer information about their physical condition and permit the early detection of health problems before the onset of evident clinical signs [4]. In fact, modifications in the short-term feeding behavior of dairy cows have been linked to the onset of disorders such as lameness or ketosis [13,14], since disease affects behavioral al motivation, resulting in decreased activity and feed intake, as well as in modifications of the feeding behavior [15].

Furthermore, rumination behavior is closely related to rumen functionality, which is negatively affected by diseases [16], heat stress [17] and proximity to stressful events, such as the onset of calving [18], resulting in a reduction in daily rumination time. Accordingly, changes in feeding and rumination time or patterns are regarded as key behavioral indicators of health issues in ruminants [19], and their monitoring is considered a useful tool for their early detection—allowing the farmer to adjust herd management—or as decision-support tool. In this framework, knowing daily feeding and rumination time becomes crucial. Nevertheless, the use of activity data from sensors to infer patterns of behavior, to adjust herd management or for the early identification of health issues requires a validation against gold standard methods, in order to assess the precision and accuracy of the measurements [4].

The primary aim of this study was to assess the agreement between feeding and rumination times in dairy cows, as measured by a commercial collar-based accelerometer, the AFICollar^®^ (Afimilk, Israel) system, and by visual observations. Further, the performance of the current version of the software, AFIfarm^®^ software 5.4 (V5.4), was compared with that of an updated version, AFIfarm^®^ 5.5 (V5.5). The possible applications of the sensor at farm-level, in relation to other sensors described in the literature, have been examined.

## 2. Materials and Methods

The experiment was carried out on the 2nd and 4th November 2020, on a commercial dairy farm located in Mantua, northern Italy (45°10′11.3″ N 10°45′01.7″ E). During the two days of trials, the temperatures ranged from 9.3 and 9.5 °C to 12.3 and 17.4 °C, with average daily RH of 100% and 95.9%, respectively (data acquired from the nearest weather station).

### 2.1. Animals and Experimental Setup

Each day, twenty Holstein-Friesian cows (778 ± 72 kg BW) were randomly divided into 4 groups and assigned to 3 feeding treatments. Animals (N = 10) in 2 out of the 4 groups received a total mixed ration (TMR); 5 animals were fed long dry hay (LHY) and 5 animals were fed grazed pasture (GRZ) with no supplementation of concentrates. During observations, TMR and LHY animals were housed in straw yard pens inside the barn, while GRZ cows remained outdoors on a ryegrass-based pasture. In TMR and LHY, fresh feed was distributed every day at 08:00 a.m. By using electing fences, the GRZ cows were offered a fresh area of pasture at the beginning of each observation period.

The TMR contained (on a DM basis) 18% corn silage, 16% grass silage, 8% grass hay, 5% alfalfa hay, 1% wheat straw, 27% ground corn, 10% soybean meal, and 15% of a commercial pelleted concentrate with 51.0% DM, 14.4% crude protein (CP), 54.0% neutral detergent fiber (NDF) and 34.5% acid detergent fiber (ADF). The LHY had 91.2% DM, 10.4% CP, 57.2% NDF and 39.0% ADF. Pre-grazing compressed grass height (measured with a rising plate meter; Grasshopper, True North Technologies, Ireland) was 78 ± 16 mm. Hand-plucked samples of GRZ (pre-grazing) contained 17.6% DM, 19.4% CP, 39.0 NDF and 28.1% ADF. In every pen as well as at pasture cows had free access to water and feed (or grass) throughout the observation periods.

### 2.2. Activity Sensors

Cow behavior was automatically recorded with the collar-based sensor system. The collar uses a 3D accelerometer positioned on the left side of the cow’s neck, which is shielded in a waterproof plastic case. In order to maintain the sensor in the desired position (on the upper left side of the cow’s neck), a weight is placed at the lowest part of the collar. The total weight of the device (including all components) was 922.4 g. To allow the cows to get used to the collar device, all animals involved were fitted with the sensor at least 14 days before the beginning of the observations. Raw accelerometer data collected by the sensor were processed using two different software versions: V5.4 and V5.5. The system returned individual hourly rumination and feeding times.

### 2.3. Behavioural Observations

Cow behavior was recorded through visual observations by 4 operators. Before the start of the experiment, an experienced instructor trained all observers on behavior classification and data recording techniques. Then, four preliminary observation sessions were carried out to assess the agreement among observers. During these sessions, the operators simultaneously observed cows in each of the four groups of animals for 20 minutes.

In the experimental sessions, each observer was assigned in rotation to one of the four pens (Table 1). Observations were carried out during three 2-hour periods (9:00–11:00 a.m., 12:00–2:00 p.m. and 4:00–6:00 p.m.), with observers positioned outside the pens (TMR, LHY) or paddocks (GRZ), in full view of all cows. Two classes of behaviors were recorded by scan sampling at 1-minute intervals, using synchronized stopwatches, according to the following ethogram: feeding (feed or grass ingestion, chewing and shallowing) and ruminating (cud regurgitation, chewing and swallowing). The dominant behavior was noted on a recording sheet at the end of every 1-minute period.

### 2.4. Data Preparation and Statistical Analysis

All data analysis was carried out using R version 3.6.1. [20]. In order to smooth behavior data and determine rumination and feeding bouts, breaks in the cow’s activity patterns (i.e., intervals between consecutive periods of the same activity) were classified as intra-bout and inter-bout intervals. As proposed by Rook and Huckle [21], intra-bouts were defined as the short breaks inside an activity period, whereas inter-bouts were defined as longer breaks between periods of the same activity. For the determination of rumination and feeding bouts, we considered the time between events that separates intervals within from intervals between bouts, in particular, the log10-frequency distribution of the interval lengths between blocks of behaviors [22,23] (hereafter, log10- interval length). In agreement with these approaches, intra- and inter-bouts were calculated by fitting a mixture of 2 Gaussian distributions to the log10- interval lengths, using the *mclust* package [24]. After the clustering procedure, all feeding and rumination intra-bouts were considered as feeding and rumination records, respectively.

Direct behavioral observations recorded at 1 min resolution were converted into hourly eating and ruminating time (min/h) to match the format of the data generated by the sensor system. In total, 240 h of behavioral observations were recorded and used for the analysis. To assess agreement among human observers, feeding and ruminating behavior data (categorical) recorded simultaneously (at 1 min resolution) by each observer were analyzed. The *irr* package [25] was used to calculate Cohen’s kappa (κ_c_) and Fleiss’ kappa (κ_f_) for pairwise and overall interrater agreement, respectively. Both κ_c_- and κ_f_-values were interpreted as follows: ≤0.20 as no agreement, 0.21–0.39 as minimal, 0.40–0.59 as weak, 0.60–0.79 as moderate, 0.80–0.90 as strong, and >0.9 as almost perfect agreement [26].

Various tests were conducted to evaluate the agreement between the sensor system and the visual observations. Hourly feeding and ruminating times (numeric) generated by both versions of the software (V5.4 and V5.5) were analyzed in comparison with the data recorded by the human observers, which were considered as the gold standard. The Spearman’s rank correlation coefficient (ρ) was computed using the package *stats* [20], while the package *epiR* [27] was used to calculate the concordance correlation coefficient (CCC). The values of ρ and CCC were considered negligible (0.00–0.30), low (0.30–0.50), moderate (0.50–0.70), high (0.70–0.90), and very high (0.90–1.00) [28].

Further, Bland–Altman plots and related statistics were obtained using the package *BlandAltmanLeh* [29]. Briefly, Bland–Altman analysis was developed to visually evaluate the agreement between two quantitative measurements by plotting the difference of the two paired measurements against the mean of the two measurements [30]. This method allows for constructing limits of agreement, which are calculated by using the mean (bias) and the standard deviation of the differences between the two measurements. In the current study, the limits of agreement were calculated as ±1.96 standard deviations from the bias. The critical difference was calculated as a 1.96 standard deviation of the difference (i.e., half the difference between the lower and upper limits of agreement). The normal distribution of the differences was visually verified. Further, as 95% CI are provided for the limits of agreement as well as for the bias, the magnitude of the systematic difference between the two measurements can be assessed [31]. A significant under- or overestimation was declared when the line of equality (mean difference = 0) was not included in the 95% CI of the bias.

In order to explore the effects of the software version and the type of feed on the sensor accuracy, a linear mixed model was fitted using the package *lme4* [32]. The difference between sensor data and visual observations (min/h) was the response variable. Separate models were fitted for the difference in feeding and ruminating time. Software version (“V5.4”; “V5.5”), type of feed (“GRZ”,”LHY”,”TMR”) and their interaction entered the models as fixed explanatory variables. A linear mixed model was also built, for both feeding and ruminating times, to assess the effect of the cow on the difference between sensor data and visual observations. In this case, the models included Cow ID, software version (“V5.4”; “V5.5”), and their interaction as fixed explanatory variables. Cow ID was included in all models as a random term. The normality and homoschedasticity of the residuals were visually evaluated. Least-squares means and pairwise comparisons (with Tukey method) were computed with the package *emmeans* [33]. Differences were considered significant at *p* < 0.05, while a tendency was declared at *p* < 0.10.

## 3. Results

Measures of interrater reliability (κ_c_ and κ_f_) are reported in Table 2. The results depict a strong overall agreement among the observers, indicating that the training procedure was adequate for the experiment.

The results of the correlation analysis between the sensor data and the human observations for feeding and rumination times are reported in Table 3. Overall, the behavioral data generated by the sensor system showed a high correlation with visual observations. With software V5.4, the sensor showed substantially better correlation statistics for feeding time than for ruminating time. The introduction of V5.5 improved the correlations of both feeding and rumination. However, the largest differences in ρ and CCC between the two software versions were found for ruminating time, which, when using V5.5, showed correlation statistics comparable to those found for feeding time.

Bland–Altman analysis allowed us to gain more insight into the agreement between the sensor data and human observations, as well as to better appreciate the differences between software versions. Bland–Altman plots for both feeding and ruminating time data generated with V5.4 and V5.5 are reported in Figure 1 and Figure 2. The sensor system with software V5.4 was found to significantly overestimate feeding time (+3.63 min/h) compared to visual observations. The development of V5.5 allowed us to reduce the sensor bias for feeding time (−1.06 min/h) to a level that was not statistically significant. Similarly, ruminating time was shown to be significantly underestimated in V5.4 (−3.93 min/h), but the sensor was capable of producing unbiased rumination data with V5.5 (−0.40 min/h).

Besides computing mean differences, the Bland–Altman plots showed that, with V5.4, the distributions of the differences between sensor data and visual observations were skewed for both feeding time (Figure 1) and ruminating time (Figure 2). With V5.5 instead, the differences appeared to be symmetrically distributed. Compared with 5.4, the software V5.5 also reduced the critical difference for feeding time as well as for ruminating time. Despite this, the limits of agreement for both feeding and ruminating times remained relatively wide, even with the software V5.5 (Table 3).

The mixed model for the differences between feeding times detected by the sensor data and by human observers highlight a significant effect of the software version. V5.5 manifested a lower mean difference compared with V5.4 (−1.21 vs. 3.05 min/h; *p* < 0.001). The main effect of type of feed did not produce statistically significant differences (2.93, −2.20 and 2.03 min/h for GRZ, LHY and TMR, respectively), even though LHY tended to be lower than GRZ (*p* = 0.092). The type of feed × software version interaction showed that software versions had significantly different effects depending on feed type (Figure 3). The software V5.5 significantly improved the sensor’s performance in detecting feeding time for both GRZ (1.05 vs. 4.82 min/h; *p* = 0.022) and TMR (−0.98 vs. 3.05 min/h; *p* < 0.001). Conversely, the new software version tended to produce a worse mean difference than V5.4 for LHY (−3.69 vs. −0.71 min/h; *p* = 0.071).

The analysis of the mixed model as regards the difference in ruminating time also showed a significant effect of software version. Overall, the development of V5.5 allowed us to significantly improve the detection of ruminating behavior compared with V5.4 (−0.42 vs. −3.44 min/h; *p* < 0.001). Type of feed alone did not affect the difference in ruminating time (−2.82, −0.35 and −2.64 min/h for GRZ, LHY and TMR, respectively), but its interaction with the software version produced significant results (Figure 4). Compared to V5.4, the software version V5.5 significantly enhanced the sensor’s accuracy in terms of TMR (−0.12 vs. −5.16 min/h; *p* < 0.001), but no differences between V5.5 and V5.4 were detected for GRZ (−1.92 vs. −3.72 min/h; *p* = 0.198) and LHY (0.77 vs. −1.46 min/h; *p* = 0.112).

The analysis of the mixed model to specifically assess inter-cow differences showed that the cow had a negligible effect on sensor performance. Although numerically some differences could be noticed among individual cows, the effect of cow ID on the differences in both feeding and ruminating times detected via the sensor data and by human observers was not statistically significant. The interaction between cow ID and software version were also non-significant for both feeding and ruminating times.

## 4. Discussion

The results of the mixed model confirm the outcomes of the agreement analysis. The adoption of the new software V5.5 led to significantly lower mean differences in both feeding and rumination times: the improvements in the algorithm for behavior classification produced a non-significant bias for V5.5, contrary to the previous version of the software V5.4. Based on the results of the agreement analysis performed on hourly data, the sensor system under software version V5.5 can be expected, on a daily basis, to underestimate feeding and ruminating times by 25.4 and 9.6 min/d, respectively. As the time budget of dairy cattle has been reported to include 2.4–8.5 h/d for feeding and 2.5–10.5 h/d for ruminating [34], the mean error introduced by the system appears rather negligible, at least from a practical point of view. Therefore, the behavioral information generated by the sensor system tested in the current study appears accurate enough to be used in common herd management applications, especially in relation to cows’ metabolic health and feeding management. Variations in feeding and ruminating times have also been associated with the onset of estrus, so the sensor system also has the potential to improve fertility management [35].

The results of the current study indicate that feeding regime, per se, did not affect the accuracy of the measures, although bias in feeding behavior tended to be lower when cows were fed with TMR and higher when grazing. As for rumination, the bias tended to be lower, but not significant, when cows were fed long hay. Kröger et al. [12] evaluated a research-grade halter sensor for rumination detection under three feeding regimes (i.e., diets differing in roughage/concentrate proportion), and also reported no effects of type of feed on the agreement between visual observation and sensor recording. This evidence may suggest that, regardless of the type of sensor (i.e., accelerometer, halter) or its position on the animal (neck, muzzle), the performances of the device in classifying rumination behavior are mildly affected by the type of feed offered.

The improvements in the accuracy of the device produced by the new software version appear evident and statistically significant when considering their interaction with the effect produced by different types of feed. Compared to software V5.4, the new version significantly increased the accuracy of measures when cows were fed TMR for both feeding and rumination times, as well as for grazing conditions, but only in relation to feeding behavior. Overall, V5.5 increased bias only when measuring time spent feeding on long hay. The passage from V5.4 to V5.5 had an inverse effect on the classification performance of feeding time in TMR and GRZ with respect to LHY.

This evidence suggests that, when fed with long hay, cows might display a different pattern of head movement, resulting in a different set of acceleration values recorded by the sensor. With both V5.4 and V5.5, bias in rumination time as well as differences between visual observations and recorded data were moderate in cows fed LHY. The increased content of fiber, and the particle size of feed in LHY compared to TMR and GRZ, are proven to affect rumination behavior in dairy cows by increasing ruminating boli per day, as well as ruminating chews per bolus [36]. With regard to these effects on chewing, emphasized head movements may have facilitated the classification of rumination in LHY treatment.

The development of the software V5.5 remarkably improved sensor accuracy and produced unbiased data; nevertheless, the relatively wide limits of agreement detected for both feeding and ruminating times indicate that sensor precision can be further improved. Compared to other collar-based accelerometers in commerce, the correlation between observed and recorded behavior was slightly lower. Grinter et al. [11] and Werner et al. [37] validated another commercially available collar-based sensor, highlighting a stronger correlation for both rumination (r = 0.99 and r = 0.97) and feeding behavior (r = 0.93 and r = 0.94). The agreement analysis performed in the current study produced bias values similar to those reported by Werner et al. [37]. The critical difference for feeding time was also similar (18.53 min/h vs. 16.29 min/h), but the critical difference for ruminating time found in the present study was considerably higher (15.40 min/h vs. 4.93 min/h).

Compared with the results of the present study, previous research on other types of sensors positioned on the cows’ head (e.g., ear tags, halters) reported higher, yet variable, correlations between visual and sensor data. For ear tags, Bikker et al. [38] reported r = 0.93 and CCC = 0.93 for rumination and r = 0.88 and CCC = 0.75 for feeding (cows fed with PMR and TMR), while Borchers et al. [10] found an r = 0.69 with CCC = 0.59 and r = 0.88 with CCC = 0.82 for rumination and feeding, respectively (cows fed with TMR). During the validation of another ear tag sensor, Borchers et al. [10] found an r = 0.96 and CCC = 0.97 for ruminating time. The position of the sensor, however, critically affects the performances of classification in relation to the target behavioral categories. Noseband sensors, overall, more efficiently detect rumination than feeding [12,37,39,40,41]. Conversely, ear tag-based sensors appear to provide better estimates of eating time [42,43]. Leg sensors are recognized as efficient tools to detect lying behavior compared to feeding [44], while accelerometer-based sensors mounted on collars appear suitable for detecting eating behavior [45,46].

The design of our study involves two TMR groups as opposed to one LHY and one GRZ. However, no differences were detected here, or in the performances of classification and in the variances among the two TMR groups. Thus, despite the imbalance in sample size across the three feeding regimes, no bias was introduced in the statistical analysis or in the results of the present study.

According to the manufacturer, one of the main goals in the development of the software V5.5 was specifically to improve the precision of the algorithm used for behavior detection. As the AFICollar^®^ is a relatively new tool and the company will continue to refine the algorithm, further improvements in the accuracy of the sensor system are to be expected in future versions of the software. Based on the results of the present study, further developments should specifically target the refinement of accuracy in feeding behavior detection.

## 5. Conclusions

The sensor system tested in the current study appears able to produce adequately reliable data on individual feeding and ruminating times. The information generated by this sensor system has the potential to improve hard management. The use of the updated software version V5.5 for behavior classification is recommended compared to the previous V5.4 release, as it enhances the sensor’s accuracy noticeably. Further developments in the classification algorithm should aim at improving precision, which currently appears to be the main limitation of this sensor system.

## Figures and Tables

**Figure 1 animals-11-02852-f001:**
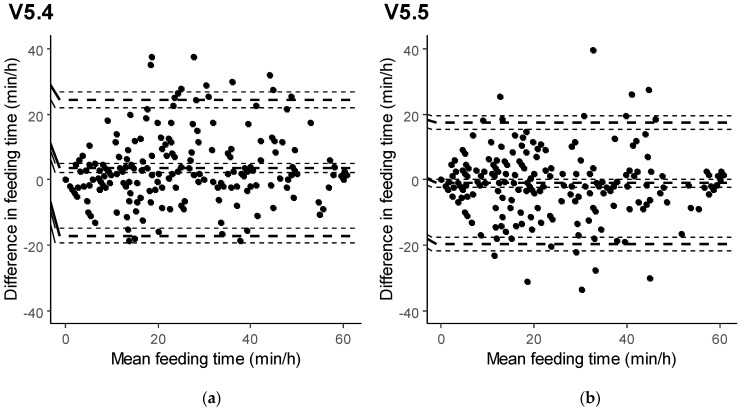
Agreement between the sensor measurements and human observations of feeding time, displayed in Bland–Altman plots (dashed lines indicate bias, upper and lower limits of agreement, each with 95% CI) for software versions 5.4 (**a**) and 5.5 (**b**).

**Figure 2 animals-11-02852-f002:**
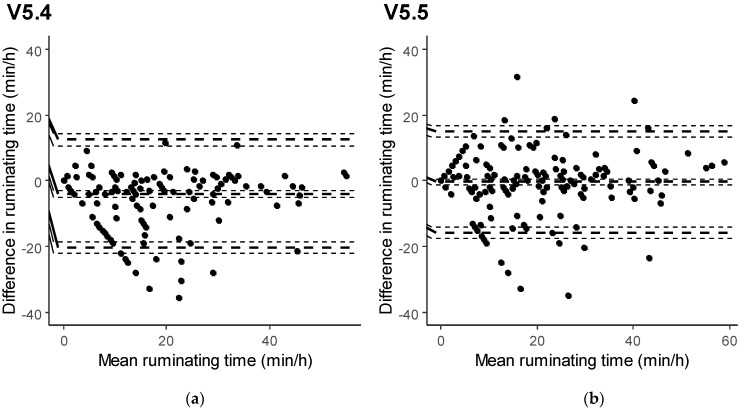
Agreement between the sensor measurements and human observations of ruminating time, displayed in Bland–Altman plots (dashed lines indicate bias, upper and lower limits of agreement, each with 95% CI) for software versions 5.4 (**a**) and 5.5 (**b**).

**Figure 3 animals-11-02852-f003:**
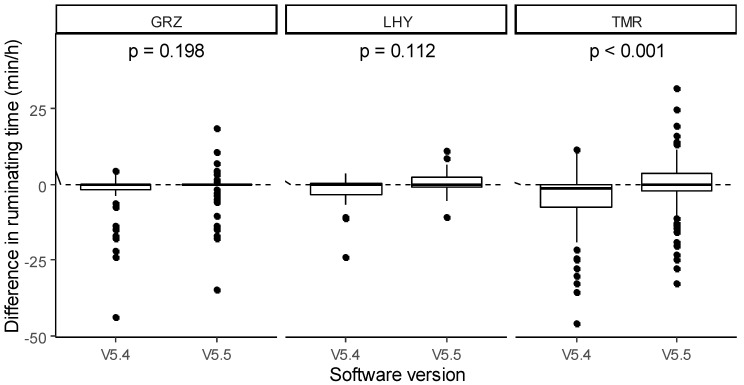
Effect of the type of feed × software version interaction on the difference in feeding time between the sensor data and human observations.

**Figure 4 animals-11-02852-f004:**
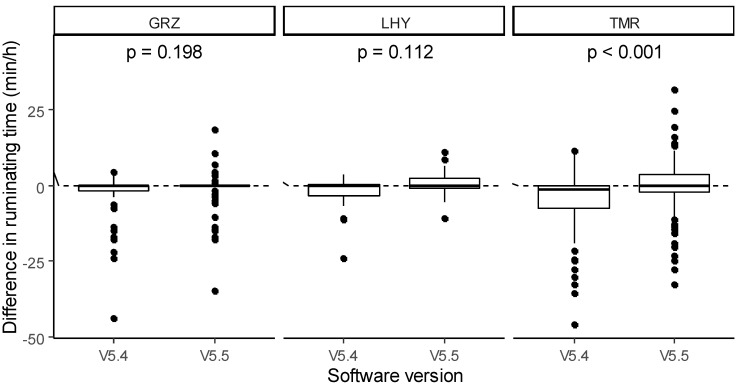
Effect of the type of feed × software version interaction on the difference in ruminating time between the sensor data and human observations.

**Table 1 animals-11-02852-t001:** Behavioral observations protocol: rotation scheme of the observers (1, 2, 3, 4) among the experimental pens (TMR1, TMR2, LHY) and pasture (GRZ).

Day	Session (Time)	Treatment			
		TMR 1	TMR 2	LHY	GRZ
Day 1	9:00–11:00 a.m.	1	2	3	4
	12:00–2:00 p.m.	2	1	4	3
	4:00–6:00 p.m.	3	4	1	2
Day 2	9:00–11:00 a.m.	1	2	4	3
	12:00–2:00 p.m.	4	3	2	1
	4:00–6:00 p.m.	2	1	3	4

**Table 2 animals-11-02852-t002:** Pairwise (Cohen’s kappa) and overall (Fleiss’ kappa) interrater reliability for the human observers visually classifying feeding and ruminating behaviors at a 1 min resolution.

Observer	1	2	3	Overall
2	0.898	-		0.881
3	0.893	0.871	-	
4	0.876	0.913	0.838	

**Table 3 animals-11-02852-t003:** The Spearman’s rho (r_s_), concordance correlation coefficient (CCC), and Bland–Altman analysis (bias, upper and lower limits of agreement with 95% CI) of automated measurements versus visual observations of feeding and ruminating behaviors.

Behavior	Software Version	r_s_	CCC	Bias (95% CI)	Lower Limit of Agreement ^1^ (95% CI)	Upper Limit of Agreement ^1^ (95% CI)	Critical Difference ^2^
Feeding time (min/h)	V5.4	0.84	0.83	3.63 * (2.27; 4.97)	−17.17 (−19.50; −14.89)	24.42 (22.08; 26.76)	20.79
	V5.5	0.85	0.86	−1.06 (−2.26; 0.14)	−19.60 (−21.68; −17.51)	17.46 (15.38; 19.55)	18.53
Ruminating time (min/h)	V5.4	0.81	0.79	−3.93 * (−4.99; −2.86)	−20.33 (−22.17; −18.49)	12.48 (10.63; 14.32)	16.41
	V5.5	0.83	0.86	−0.40 (−1.40; 0.60)	−15.80 (−17.53; −14.07)	15.00 (13.27; 16.73)	15.40

^1^ Limits of agreement ware calculated as ±1.96 standard deviations from the bias. ^2^ Critical difference was calculated as 1.96 standard deviations of the differences. * Bias between sensor data and visual observations significant at 95%.

## Data Availability

Restrictions apply to the availability of these data. Data were obtained from the farm Leso Bruno Massimo e Tiziano s.s. (Mantova, Italy) and Afimilk (Kibbutz Afikim, Isarael). Data are available from the authors with the permission of these companies.

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
