# Peer review of "Validation of a Commercial Collar-Based Sensor for Monitoring Eating and Ruminating Behaviour of Dairy Cows"

_animals, 2021, doi:10.3390/ani11102852_

Round 1
Reviewer 1 Report
Dear Authors,
I'm very interested by this topic. I found your study very interesting. I have made comments in your text, in annex.
Best regards,

Author Response
Line 14: It looks this comment (on the pdf) is empty
Line 21 (here and throughout): Ok, edited all ® (registered trademark), now as superscript
Line 53: Ok, definition of RFID added
Line 70: Ok, now “behavioural”
Line 76: Ok, now 1 word “behavioural”
Line 92: Ok, edited “nd” and “th”, now as superscript
Line 93: Ok, added GPS coordinates
Line 103-1: Ok, added a tense about time of feed distribution in TMR and LHY
Line 103-2: Yes, every day a fresh grass allocation before each of the 3 observation periods. We don’t feel the need to specify this at the number and time of the periods have been already described (table 1).
Line 108-1: Ok, added %
Line 108-2: Ok, now “compressed”
Line 109: No, we did not measure/estimated grass DMI (nor the DMI of other groups TMR and LHY) but we acknowledge this information could have been of some interest. That said, we don’t think this is a major limitation for this kind of study, which focuses on validating the sensor system. For this reason, we prefer not to mention (grass) DMI in this manuscript.
Line 115: Yes, added a tense about the adaptation period.
Lines 144-148: The log10 transformation was applied to interval lengths between behaviours following the approach described first by Tolkamp et al. (1998), and applied also by De Vries et al. (2003). Tolkamp et al. (1998) proposed as criterion for meal definition an approach based on plotting the frequency distribution of log10-transformed intervals between meals and using discontinuities in the distribution to determine objectively which intervals were within meals and which were beetween meals.
Line 149: Two Gaussian distributions were fitted for each behaviour, one to detect intra-bouts and the other to detect inter-bouts.
Line 201: Yes, there were differences in the “total” feeding and ruminating times observed during the different periods. That was largely expected as cows are known to be more active (especially for feeding/grazing) during certain parts of the day. Even though this might be interesting from an ethological stand point (but already very well documented) it has little to do (at least in our opinion) with the assessment of the sensor performance or agreement. In this case we focused on the difference between the sensor data vs observers’ data that is independent from the total time spent doing one activity or the other during the observation periods. In fact, we did not detect any effect of period (or time of the day) on the difference between observer and sensor data. We feel that adding a discussion about this could create confusion to the reader. So if you agree we would avoid discussing this specifically in the paper.
Table 3: See response to comment L201 about effect of observation periods. Regarding inter-cow differences, during the analysis we actually tested for the effect of individual cow (cow ID) on the difference between observers and sensor data. As we didn’t detect any significant effect we initially decided not to report these results. However, we recon this could be of interest in the context of this study so we now added tenses about effect of cow both in the methods (section 2.4) and in the results.
Line 245: So as you noticed, there are no differences in the feeding/treatment between the two TMR groups but we still had two of them. We had to do it for practical reasons even if that resulted in an “unbalanced” study design. Basically, we wanted to test the sensor under 3 feeding regimes but we had 4 observers available for the study. Obviously we liked to use all the workforce available to maximise data collection over the time we were allowed to use the cows for the experiment. As we found that one observer can effectively monitor only 5 animals at a time, to reach 20 cows in total we decided to have 2 TMR groups, which was thought to be the most representative/important feeding regime. Now, we reconize the unbalanced sample size across the feeding regimes could limit the power of the test but we don’t think this introduces “bias” nor violates the usual assumptions as we didn’t notice appreciable differences in the variance within the groups. However, the issue was addressed in the final part of the discussion paragraph.
Line 270: See response to comment for line 245
Line 279: Comments about it were added in the discussion
Line 313: Ok, cows feed regimes were included.
Line 347: Capital letter added
Line 355: Capital letter added
Line 383: Sorry, we do not understand which capital lettrers should be changed
Reviewer 2 Report
The paper validates a commercial collar-based sensor for the classification of cattle behaviours (ruminating and eating). The study evaluates the performance of the system for different software versions (AFIfarm® V5.4 and V5.5) and different types of feed (total mixed ration, long hay, grazed grass). The paper is well written and clear. However, some points should to be addressed in order to improve the paper quality before considering for publishing:
- Line 54: add references to each technology (eartags, bolus, etc.)
- Line 57: “The latter are the most common” add reference.
- Lines 116-117: “In order to maintain the sensor in the desired position, a weight is placed at the lowest part of the collar”
Is there a recommendation from the provider on which position is the best (top, side, right left...)? How the results/performance would change if this position is changed over time?
- Is there information about the sampling rate of the accelerometer?
- Is the data processed locally, transited to a backend system in real-time or the data is stored locally and processed afterwards by the software? Clarify this in section 2.2.
- What is the expected lifetime of the system?
- What about the missed data?
- Line 208: what is the expected daily error? The daily feeding time is 3-5 hours while ruminating time is 9-12 hours.
- Line 307: Correct the values 15.40 min/h vs 4.93 min/h.
- What are the implications of these results in practice (detection of heat, lameness)? Are these results sufficient for such applications/other application?
- Other studies should be cited as well:
- Daniel Smith et al. 2016, “Behavior classification of cows fitted with motion collars: Decomposing multi-class classification into a set of binary problems” Computers and Electronics in Agriculture
- Said Benaissa et al. “Classification of ingestive-related cow behaviours using RumiWatch halter and neck-mounted accelerometers” Applied Animal Behaviour Science
- Arcidiacono et al. 2017, “Development of a threshold-based classifier for real-time recognition of cow feeding and standing behavioural activities from accelerometer data” Computers and Electronics in Agriculture
- Vázquez Diosdado et al. 2015, “Classification of behaviour in housed dairy cows using an accelerometer-based activity monitoring system” Animal Biotelemetry
- The comparison to other positions (ear, leg, noseband) should be detailed further in the discussion with clear recommendations on which position is the best, for which behaviour.
- Add the study limitations.
- What are the future steps after the validation?
Author Response
Line 54: add references to each technology (eartags, bolus, etc.)
AU: References were added for the cited technologies
Line 57: “The latter are the most common” add reference.
AU: Reference added
Lines 116-117: “In order to maintain the sensor in the desired position, a weight is placed at the lowest part of the collar ”Is there a recommendation from the provider on which position is the best (top, side, right left...)? How the results/performance would change if this position is changed over time?
AU: Ok, added “(on the upper left side of the cow’s neck)”. Actually we don’t know what happens if the sensor is misplaced or if the position changes as we strived to fit all collars properly (according to the manufacturer’s recommendation).
Is there information about the sampling rate of the accelerometer? Is the data processed locally, transited to a backend system in real-time or the data is stored locally and processed afterwards by the software? Clarify this in section 2.2. What is the expected lifetime of the system?
AU: Yes, we understand some more details about the sensor are missing. Unfortunately, the manufacturer doesn’t allow us to share technical details so we are unable to answer these questions.
What about the missed data?
AU: Sorry, we didn’t really understand this comment. We didn’t have missing data in this study.
Line 208: what is the expected daily error? The daily feeding time is 3-5 hours while ruminating time is 9-12 hours.
AU: Ok, we added a paragraph about this in the discussion. The best reference we found (https://doi.org/10.3168/jds.2017-13706) reported ranges of 2.4–8.5 h/d for feeding and 2.5–10.5 h/d for ruminating (yes, they look very wide also to us) so this is what we also reported in the manuscript.
Line 307: Correct the values 15.40 min/h vs 4.93 min/h.
AU: Ok, edited with actual values for feeding, now (18.53 min/h vs 16.29 min/h)
What are the implications of these results in practice (detection of heat, lameness)? Are these results sufficient for such applications/other application?
AU: Ok, we expanded the discussion about this in the discussion section
Other studies should be cited as well:
Daniel Smith et al. 2016, “Behavior classification of cows fitted with motion collars: Decomposing multi-class classification into a set of binary problems” Computers and Electronics in Agriculture
Said Benaissa et al. “Classification of ingestive-related cow behaviours using RumiWatch halter and neck-mounted accelerometers” Applied Animal Behaviour Science
Arcidiacono et al. 2017, “Development of a threshold-based classifier for real-time recognition of cow feeding and standing behavioural activities from accelerometer data” Computers and Electronics in Agriculture
Vázquez Diosdado et al. 2015, “Classification of behaviour in housed dairy cows using an accelerometer-based activity monitoring system” Animal Biotelemetry
AU: We included more studies in the manuscript, also those from Smith et al., Benaissa et al. and Diosdado et al.
The comparison to other positions (ear, leg, noseband) should be detailed further in the discussion with clear recommendations on which position is the best, for which behaviour.
AU: Ok, these information were added in the discussion
Add the study limitations.
AU: Study limitations were added in the final part of the discussion
What are the future steps after the validation?
AU: We added recommendations for further developments at the end of the discussion section.
Reviewer 3 Report
Overall comments:
Dear authors, nice study. I wonder why the company implemented a new version of this software. Can you point out if this change was aimed specifically to improve accuracy of detecting feeding behaviours? I ask this because if the version 5.5 wasn’t aimed specifically at improving these measurements could we assume that version 5.6 and onwards would be better than version 5.5 for feeding related variables or could it be that a newer algorithm would perform worse? Some mention to it in your discussion would raise awareness for future researchers that may end up using this system for their research.
Specific comments
L9: “Io”?
L23: Specify how many groups.
L24: Not sure what you mean by grazed grass, maybe “animals were allowed to graze”?
L25: Specify if continuous or scan sampling observation.
L76: Behavior al
L129: as min/h?
Author Response
Dear authors, nice study. I wonder why the company implemented a new version of this software. Can you point out if this change was aimed specifically to improve accuracy of detecting feeding behaviours? I ask this because if the version 5.5 wasn’t aimed specifically at improving these measurements could we assume that version 5.6 and onwards would be better than version 5.5 for feeding related variables or could it be that a newer algorithm would perform worse? Some mention to it in your discussion would raise awareness for future researchers that may end up using this system for their research
AU: Yes, we added a comment about this at the end of the discussion section.
L9: “Io”?
AU: Ok, now “In order”
L23: Specify how many groups.
AU: Ok, added “4”
L24: Not sure what you mean by grazed grass, maybe “animals were allowed to graze”?
AU: Ok, now “animals allowed to graze”
L25: Specify if continuous or scan sampling observation.
AU: Ok, added “scan sampling at 1-minute interval"
L76: Behavior al
AU: Ok, now “behavioural”
L129: as min/h?
AU: Observations generated by the observers were recorded (by scan sampling at 1-minute intervals) as categorical values. For the agreement among observers we used those categorical values (Cohen’s k) while, for the agreement between observations and sensor, the data were converted in min/h (numeric). This is already described in section 2.4 “Data preparation and statistical analysis” so we don’t feel the need to add text about it here.
Round 2
Reviewer 2 Report
The authors addressed all my comments. Thanks!